# Research on the Application and Performance Optimization of GPU Parallel Computing in Concrete Temperature Control Simulation

Xuerui Zheng [1], Jiping Jin [2], Yajun Wang [2], Min Yuan [1] and Sheng Qiang [1,*]

1   College of Water Conservancy and Hydropower Engineering, Hohai University, Nanjing 210098, China; zxrchinahohai@hhu.edu.cn (X.Z.); miny@hhu.edu.cn (M.Y.)
2   The First Engineering Bureau of Henan Water Conservancy, Zhengzhou 450016, China; jinjiping12@163.com (J.J.); 110wyj@163.com (Y.W.)
*   Correspondence: sqiang2118@hhu.edu.cn

**Abstract:** With the development of engineering technology, engineering has higher requirements for the accuracy and the scale of simulation calculation. The computational efficiency of traditional serial programs cannot meet the requirements of engineering. Therefore, reducing the calculation time of the temperature control simulation program has important engineering significance for real-time simulation of temperature field and stress field, and then adopting more reasonable temperature control and crack prevention measures. GPU parallel computing is introduced into the temperature control simulation program of massive concrete to solve this problem and the optimization is carried out. Considering factors such as GPU clock rate, number of cores, parallel overhead and Parallel Region, the improved GPU parallel algorithm analysis indicator formula is proposed. It makes up for the shortcomings of traditional formulas that focus only on time. According to this formula, when there are enough threads, the parallel effect is limited by the size of the parallel domain, and when the parallel domain is large enough, the efficiency is limited by the parallel overhead and the clock rate. This paper studies the optimal Kernel execution configuration. Shared memory is utilized to improve memory access efficiency by 155%. After solving the problem of bank conflicts, an accelerate rate of $437.5\times$ was realized in the subroutine of the matrix transpose of the solver. The asynchronous parallel of data access and logical operation is realized on GPU by using CUDA Stream, which can overlap part of the data access time. On the basis of GPU parallelism, asynchronous parallelism can double the computing efficiency. Compared with the serial program, the accelerate rate of inner product matrix multiplication of the GPU asynchronous parallel program is $61.42\times$. This study further proposed a theoretical formula of data access overlap rate to guide the selection of the number of CUDA streams to achieve the optimal computing conditions. The GPU parallel program compiled and optimized by the CUDA Fortran platform can effectively improve the computational efficiency of the simulation program for concrete temperature control, and better serve engineering computing.

**Keywords:** concrete; simulation computing; GPU; parallel computing; CUDA Fortran; shared memory; asynchronous parallel



## 1. Introduction

In the finite element calculation of temperature field and stress field of hydraulic concrete structures [1–3], especially the concrete structure with cooling water pipe [4], in order to get high accuracy [5,6], the calculators usually build a dense model, which leads to the increase of calculation scale [7]. In addition, reasonable temperature control calculation should be carried out in real time with the construction [8,9], so that the boundary conditions and parameters can be continuously adjusted and corrected with reference to the measured data, so as to obtain more reasonable results. Due to the low computational efficiency of the traditional serial program, it cannot meet the actual needs of engineering under the condition of ensuring the calculation accuracy. All the above

problems show that the existing serial simulation calculation method needs comprehensive improvement to improve the computational efficiency.

In recent years, the growth of the CPU clock rate has been slowing down year by year. In the field of engineering computing [10], the exploration of improving computing efficiency is gradually transferred to the parallelization implementation of programs [11]. Many explorations have been made for the implementation of multi-core CPU parallelism. Mao parallelized the joint optimization scheduling of multiple reservoirs in the upper reaches of Huaihe River, which took 5671.1 s for 1 CPU core and 2104.6 s for 6 CPU cores, with an accelerate rate of 2.704, and an efficiency of 45% [12]. Jia proposed a master-slave parallel MEC based on MPI, and analyzed the effects of task allocation, communication overhead, sub-population size, individual evaluation time and the number of processors on the parallel speedup [13]. CPU parallelism relies on adding more CPU cores to achieve speedup. Whether buying a multicore CPU or cluster parallelism [14], an order of magnitude increase in the number of CPU cores is definitely very expensive. The GPU of a home grade graphics card contains hundreds or thousands of processing cores [15–17].

The full name of GPU is the graphics processing unit, which is designed for image computing, and the research in the field of image computing has been very mature. As a fine-grained parallel method, GPU was designed for compute-intensive, highly parallel computing, which enabled more transistors to be used for data processing, rather than data caching or flow control. GPU has an absolute advantage over CPU in the number of processors and threads. Therefore, GPU computing is very suitable for the field that requires large-scale parallel computing [18]. In 2009, Portland Group (PGI) and NVDIA jointly launched the CUDA Fortran compiler platform, which greatly expanded the ication range of GPU general purpose computing. The release of this platform made GPU computing widely used in medicine, meteorology, fluid calculation, hydrological prediction and other numerical computing fields [19–22]. T Mcgraw presented a Bayesian formulation of the fiber model which showed that the inversion method can be used to construct plausible connectivity [23]. An implementation of this fiber model on the graphics processing unit (GPU) is presented. Qin proposed a fast 3D registration technique based on CUDA architecture, which improves the speed by an order of magnitude while maintaining the registration accuracy, which is very suitable for medical clinical ications [24]. Takashi uses GPU to implement ASUCA, the next generation weather prediction model developed by the Japan Meteorological Agency, and achieves significant performance speedup [25]. Taking pres-tack time migration and Gazdag depth migration in seismic data processing as a starting point, Liu introduced the idea, architecture and coding environment of GPU and CPU co-processing with CUDA [26]. GPU acceleration for OpenFMO, a fragment molecular orbital calculation program, has been implemented and its performance was examined. The GPU-accelerated program shows 3.3× speedups from CPU. MA Otaduy presented a parallel molecular dynamics algorithm for on-board multi-GPU architectures, parallelizing a state-of-the-art molecular dynamics algorithm at two levels [27]. Liang tried the GPU platform and FDTD method to solve Maxwell equations with complex boundary conditions, large amount of data and low data correlation [28]. Wen presented a new particle-based SPH fluid simulation method based completely on GPU. By this method, a hash-based uniform grid is constructed firstly on GPU to locate the neighbor particles faster in arbitrary scale scenes [29]. Lin presented a new node reordering method to optimize the bandwidth of sparse matrices, resulting in a reduction of the communication between GPUs [30]. JC Kalita presented an optimization strategy for the BiCGStab iterative solver on GPU for computing incompressible viscous flows governed by the unsteady N-S equations on a CUDA platform [31]. Tran harnessed the power of accelerators such as graphics processing units (GPUs) to accelerate numerical simulations up to 23× times faster [32]. Cohen used a GPU-based sparse matrix solver to improve the solving speed of the flood forecasting model, which is of great benefit to the early warning and operation management during the flood [33]. Ralf used CUDA to numerically solve the equations of motion of a table

tennis ball and performed statistical analysis of the data generated by the simulation [34]. The Mike (Mike11, Mike21 and Mike3) series of commercial software developed by the Danish Institute of Hydraulics has also introduced GPUs to improve the computational efficiency of models [35–37].

GPU parallel computing is rarely used in the field of concrete temperature and stress field simulation. Concrete simulation calculation is to simulate the construction process [38], environmental conditions [39], material property changes and crack prevention measures and other factors as accurately as possible. It is of great significance to GPU parallel computing to concrete temperature control simulation calculation for real-time inversion of temperature field and stress field and shortening calculation time. In this paper, the CUDA Fortran compiler platform is used to transform the large massive concrete temperature control simulation program into the GPU parallel program. An improved analysis formula of the GPU parallel algorithm is proposed. Aiming at the extra time consumption of GPU parallel in the indicators, two measures are proposed to optimize the GPU parallel program: one is to use shared memory to reduce the date access time, and the other is to hide the date access time through asynchronous parallelism.

## 2. Improved Analytical Formula for GPU Parallel Algorithms

The computation time of GPU parallel programs is affected by many factors. In order to better analyze and measure the advantages and disadvantages of parallel algorithms, a new analysis indicator based on the traditional parallel algorithm speedup is proposed. Through this theory, the relationship between the various elements of parallel computing can be better analyzed, so as to find the factors that restrict the improvement of parallel efficiency, and better serve the optimization of efficient GPU parallel computing programs.

### 2.1. CUDA Fortran Parallel Experimental Platform

The program is implemented in PGI compiler by the CUDA Fortran compiler platform [40]. The parameters of the experimental platform are shown in Table 1.

**Table 1.** Parameters of experimental platform.

| Platform | GPU | Compute Capability | Total Global (MiB) | Bus Width | Clock Rate | Processors | SM | Memory Copy | Rate (Gib/s) |
|---|---|---|---|---|---|---|---|---|---|
| 1 | GTX1050 | 6.1 | 2048 | 128 bits | 1455 MHz | 640 | 5 | H2D<br>D2H | 11.33<br>11.32 |
| 2 | GTX1070 | 6.1 | 4096 | 256 bits | 1759 MHz | 1920 | 15 | H2D<br>D2H | 11.96<br>11.73 |
| 3 | RTX3080 | 8.6 | 4096 | 256 bits | 1770 MHz | 8960 | 70 | H2D<br>D2H | 23.97<br>23.71 |

Windows10 64x.Inteli7-12700, Visual Studio 2010 PGI Visual Fortran, CUDA Toolkit 10.0.

### 2.2. The Traditional Analytical Formula

The only indicator to evaluate the computational efficiency of traditional serial programs is the computation execution time. The formula for calculating the accelerate rate of the traditional parallel algorithm is as follows:

$$R(n) = \frac{T_0}{T_1(n)} \tag{1}$$

where $R(n)$ is an accelerate rate, $T_0$ is the serial computation running time, $n$ is the number of processors, $T_1(n)$ is the parallel execution time using $n$ processors. $R(n) \in (0, n]$. The only index of the formula is time; although the formula can objectively reflect the acceleration effect of the calculation program, it cannot reflect the various factors that affect the calculation time.

*2.3. Improved Analytical Formula*

During the execution of a program, not all processes and commands can be merged. The speedup of the code is limited by the proportion of the whole program that can be parallelized. In view of this, this paper divides the computation time of the program into serial domain $T_s$ and parallel domain $T_p(n)$.

$$T_p(n) = \frac{T_0 - T_s}{n} \tag{2}$$

where $T_p(n)$ is parallel time for execution by $n$ processors in parallel domain, $T_s$ is the serial time by a single processor in serial domain. The total time the program takes to run is

$$T_1(n) = T_p(n) + T_s \tag{3}$$

Insert Equations (2) and (3) into Equation (1), new parallel computing accelerate rate:

$$R(n) = \frac{T_0}{T_1(n)} = \frac{T_0}{\frac{T_0 - T_s}{n} + T_s} = \frac{1}{\frac{T_0 - T_s}{T_0 n} + \frac{T_s}{T_0}} = \frac{1}{\frac{1}{n} - \frac{1}{n}\frac{T_s}{T_0} + \frac{T_s}{T_0}}$$
$$R(n) = \frac{1}{\frac{1}{n} + \frac{T_s}{T_0}\left(1 - \frac{1}{n}\right)} \tag{4}$$

The above formula is the speed ratio of parallel computation under ideal conditions, but the execution of the GPU parallel code also causes some other time-consuming events. Firstly, compared with the CPU parallel computing, it also costs time to transfer data between the Host and the Device (H2D and D2H). Secondly, if the processors are not allocated the same amount of work, the load will be unbalanced, and some threads will be idle, which will affect the execution speed. Finally, there is a time penalty for calling the GPU Kernel subroutine. To investigate the effects of these costs attributed to parallel computing, we define these extra time costs as $T_e$.

At present, the computing power of a GPU single core is lower than that of a CPU single core, so the attenuation coefficient $\lambda$ of GPU computing power is used to represent the ratio of GPU and CPU computing power. The actual operation time $T_G(n)$ of computing the parallel domain by the GPU core is obtained as follows:

$$T_G(n) = \lambda T_p(n) + T_e \tag{5}$$

Replace $T_1(n)$ in Equation (4) with $T_G(n)$, the total time of parallel operations at the GPU core, and insert Equation (2) into Equation (4). After collation, the parallel accelerate rate calculation formula considering computing power attenuation and extra time costs is as follows.

$$R(n) = \frac{T_0}{T_G(n)} = \frac{T_0}{\lambda T_p(n) + T_e} = \frac{T_0}{\lambda \frac{T_0 - T_s}{n} + T_e} = \frac{1}{\lambda \frac{T_0 - T_s}{T_0 n} + \frac{T_e}{T_0}}$$
$$R(n) = \frac{T_0}{T_G(n)} = \frac{1}{\frac{\lambda}{n} + \frac{T_s}{T_0}\left(1 - \frac{\lambda}{n}\right) + \frac{T_e}{T_0}} \tag{6}$$

Formula (6) makes up for the shortcomings of the traditional calculation formula that only focuses on the calculation time, considering the accelerate rate $R(n)$ of parallel computing, the core clock rate loss $\lambda$ of GPU, the number of processors $n$, the extra time cost $T_e$ and the proportion of serial domain $T_s$.

In general, the speedup increases with the number of threads $n$ participating in the computation for the same task, but the rate of increase slows down gradually due to the increase in communication overhead between threads. As $\frac{\lambda}{n}$ approaches 0, which means that GPU computing power is infinite, $R(n) = \frac{T_0}{T_s + T_e}$. The speedup of the program is restricted by the serial duration $T_s$ and the extra time cost $T_e$, and the value eventually tends to a constant $\frac{T_0}{T_s + T_e}$. Similarly, as $T_s$ approaches 0, the speedup of the program is restricted by the serial duration $\frac{\lambda}{n}$ and the extra time cost $T_e$. The value of the speedup ratio is proportional to $n$ and inversely proportional to $T_e$. In addition, the speedup will increase

if the computational part of a task is larger. This is because the communication overhead $T_e$ at the device side and the host side is reduced in proportion to the computation time.

The acceleration effect of the program is restricted by many conditions. The more cores participate in the calculation, the larger the proportion of parallel fields, the smaller the attenuation of computing power and the less additional consumption, the more computationally efficient the program is. According to the improved parallel analytical formula (Equation (6)) and the above analysis, it can be seen that in order to obtain a better acceleration effect of the GPU parallel program, we should pay attention to the following aspects: 1. Using the GPU core with more computing power and more frequency to reduce the attenuation coefficient $\lambda$ in GPU parallel computing. 2. Using more cores of the GPU to increase the value of $n$ to reduce the operation time. 3. By improving the problem solving algorithm and parallel program, the proportion of parallel operation can be increased and the extra cost of parallel program $T_e$ can be reduced.

Aspects 1 and 2 are hardware level optimization; this paper mainly focuses on the third point, the optimization of parallel computing programs.

## 3. Research on GPU Memory Access Optimization by Using Shared Memory Results

### 3.1. Selection of Kernel Execute Parameter

The main program implements GPU parallel computing through ''call Kernel <<<Dg,Db >>> (parameter list)" to write on the GPU device. Inside the <<< >>> operational character is the execution parameter of the kernel function. The parameter Dg is used to define the arrangement and number of blocks. The parameter Db is used to define the arrangement and number of threads in a block. NVIDIA uses 32 threads to form a warp, and threads in a Warp must be in the same block. Warp is the basic unit of scheduling and execution. If the number of threads in the block is not a multiple of the Warp size, the system will automatically fill the number of threads to a multiple of 32, and the extra threads are inactive threads.

It can be seen from the Figure 1 that the increase of the total number of threads in the calculation process can effectively shorten the calculation time. When the thread increment is not an integer multiple of 32, the running time will first increase and then decrease. When the thread increment is too small, the number of parallel processes in a Warp increases less, and the number of inactive threads accounts for a large number. For example, when the total number of threads is increased by 1, the resource occupancy of 32 threads is actually increased, but only one active thread provides the actual computing power, and the inactive thread occupies many SM resources not to provide speedup. As the proportion of active threads increases, the speedup effect provided by the additional threads will outweigh the negative impact of inactive threads. The whole program presents a speedup effect. Therefore, we should set the number of threads to match the characteristics of Warp, to be an integer multiple of 32.

We take the GeForce GTX1050 graphics card as an example to study the data processing bandwidth under different execution parameters, Dg(blocks) and Db(threads/block), with a consistent total number of threads, and the results are shown in Table 2.

**Table 2.** Kernel bandwidth under different execution parameters.

| Blocks | Threads/Block | Bandwidth (GB/s) | Blocks | Threads/Block | Bandwidth (GB/s) |
|--------|---------------|------------------|--------|---------------|------------------|
| 64 | 32 | 67.52 | 8 | 256 | 90.83 |
| 32 | 64 | 87.69 | 4 | 512 | 89.23 |
| 16 | 128 | 89.76 | 2 | 1024 | 79.32 |

For NVIDIA GeForce series graphics cards, max threads per multiprocessor is 2048. blocknumber = Max Threads per Multiprocessor/Thread number. As the number of threads increases, the number of blocks decreases. From the table, the speedup effect first increases and then decreases with the increase of the number of threads per SM(Multiprocessor). When the number of threads is 32, the number of blocks is large, and the resources of

SM are mainly used for the scheduling of blocks. Although this configuration can hide the scheduling delay of SM to a large extent, a block only dominates 32 threads for data operation, and there is a large gap between the data computing capacity and the peak capacity. Similarly, when the number of threads is 1024, SM only processes 2 blocks, although the computing capacity is sufficient, it cannot achieve the effect of hidden sm scheduling, and there will be a partial loss of computing capacity. The computing power of this configuration is not far from the peak capacity. The optimal number of threads is 256.

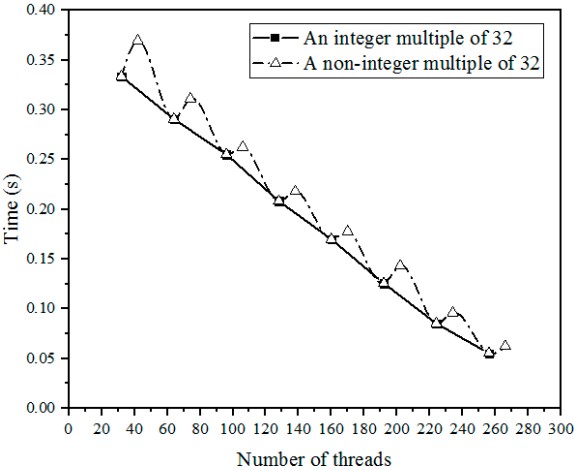

**Figure 1.** Calculation time for different number of threads.

*3.2. Features of Shared Memory*

The architecture of GPU memory is similar to that of CPU memory. In order to adapt to graphics calculation, the graphics card has made a special design and partitioned the video memory in more detail. Include: local memory, global memory, shared memory, register, L1/L2 Cache and other memory. As following Figure 2 shows:

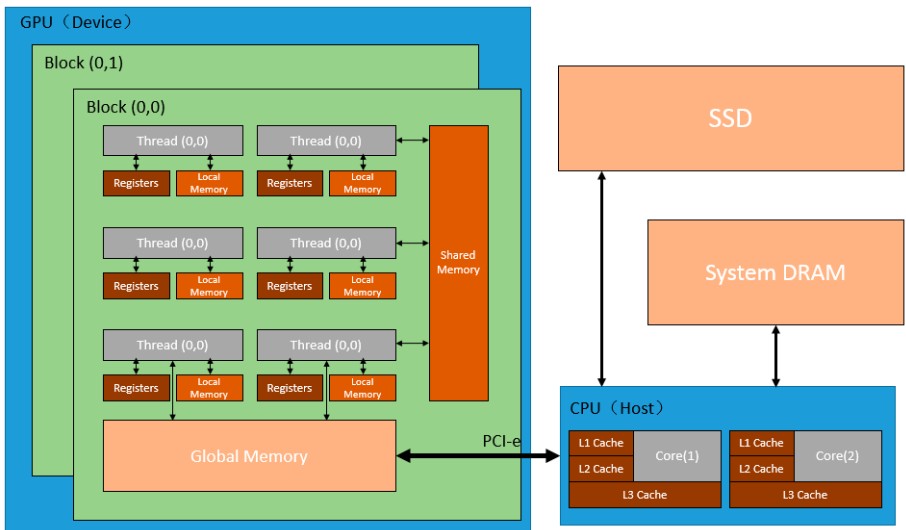

**Figure 2.** Simplified host and device memory architecture.

Different memory has different characteristics; one needs to choose the appropriate memory according to the requirements of the program variables. The register is the cache on the GPU chip, which is the fastest memory in the GPU. Same as registers, shared memory is located on the GPU chip. It is an efficient memory for thread collaboration. All threads in the same block can access variables in shared memory. The use of shared memory requires manual variable declarations in the Kernel: _device_,_shared_, for example: real(kind = 8),

shared::psum(*).The location and speed of L1/L2 cache and shared memory are very similar. The difference is that the use of L1/L2 cache is controlled by the system, and the use of shared memory is controlled by the user. The local memory is also the memory on the GPU chip, but is only retrieved by the thread. Global memory is implemented by Dynamic Random Access Memory (DRAM). It is an independent off-chip memory, which is commonly referred to as video memory. It can be accessed by all threads on the device and shared globally.

According to the characteristics of each memory and combining Figure 2 and Table 3, we can see that the fastest memory that a user can actively allocate is shared memory. Shared memory is better for exchanging data between threads than local memory. For the data that needs to be read and written repeatedly, it can be stored on the shared memory. Compared with the memory access in global memory, data access operations directly on shared memory can effectively improve the memory access efficiency of GPU.

**Table 3.** GPU memory characteristics.

| Memory | Sphere of Action | Life Cycle | Speed Ordering | Control |
|---|---|---|---|---|
| Register | Thread | kernel | 1 | System |
| L1/L2 Cache | SM | kernel | 2 | System |
| Shared Memory | Block | kernel | 2 | User |
| Local Memory | Thread | kernel | 2 | User |
| Global Memory | Gride | Program | 3 | User |

*3.3. Research on Avoiding Bank Conflicts*

Bank Conflict is a concept used for shared memory, which means that a thread has a conflict in accessing a shared memory bank. For example, two or more threads access a bank at the same time. Consider a two-dimensional B array of size 4 × 32. This array B is data processed by threads in one warp.

As can be seen from Figure 3b, threads in a warp access the first element of their own vector. The thread index is ThreadIdx.x. The access data address of each thread is (ThreadIdx.x+0) × 32. The address of the bank is ThreadIdx.x/4+0. The thread index indicates that a bank conflict will occur. Multiple threads accessing different data in the same bank, and the access of a thread to data needs to wait for the completion of the access of the previous thread, as shown in Figure 4a.

**Figure 3.** The original arrangement of the data. (**a**) Data in shared memory. (**b**) Data mapped onto the bank.

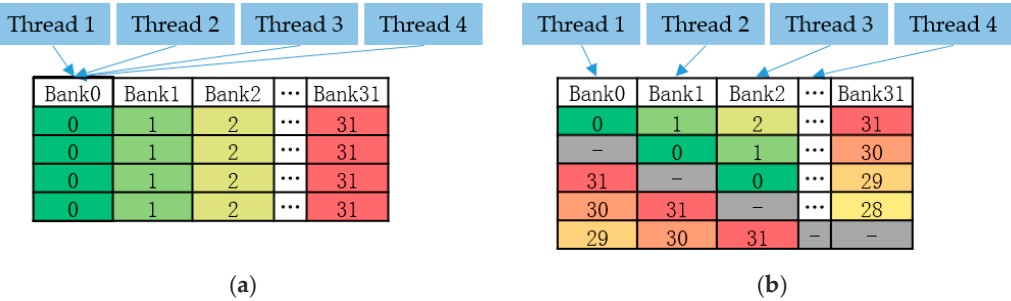

(**a**)  (**b**)

**Figure 4.** Thread data access on bank. (**a**) The original matrix. (**b**) The augmented matrix.

In order to solve the bank conflict, we propose a method of increasing the size of data stored in shared memory to solve it. The specific operation is to store one more column of empty data, as shown in Figure 5a. This can realize the misalignment of data stored in the bank as shown in Figure 5b. The code is shown in Listing 1.

(a)    (b)

**Figure 5.** The augmented arrangement of the data. (**a**) Data in shared memory. (**b**) Data mapped onto the bank.

**Listing 1.** Code of shared memory.

```
1   attributes(global) subroutine NoBF_Kernel(odata, idata)
2       real, intent(out)::odata(ny, nx)
3       real, intent(in)::idata(nx, ny)
4       . . .
5       real, shared::tile(TILE_DIM, TILE_DIM)        ! Before optimization
6       real, shared::tile(TILE_DIM, TILE_DIM + 1)    ! After optimization
7       integer::x, y, j
8       . . .
9   end subroutine
```

As can be seen from Figure 5b, threads in a warp access the first element of their own vector. The thread index is ThreadIdx.x. The access data address of each thread is ThreadIdx.x×32+x. The address of the bank is ThreadIdx.x/4+x. The data accessed by each thread is distributed in different banks, and the access of threads to banks is malposed. This method avoids bank conflicts. The disadvantage of this method is that it takes up an extra column of shared memory. If a Kernel is overloaded with shared memory, this method is not available.

*3.4. Time Consumption Analysis of Shared Memory*

For a 1024 × 1024 matrix, the computation time for matrix addition and transpose is in Table 4:

**Table 4.** Time consumption for matrix operations.

| Graphics Card | Version | | | | |
|---|---|---|---|---|---|
| | Replication | Replication (Shared Memory) | Transposition | Transposition (Shared Memory) | Transposition (Non-Bank Conflict) |
| Time (ms) | | | | | |
| Inteli7-6700 | 4974.03 | - | - | - | - |
| GTX 1050 | 91.23 | 88.31 | 336.38 | 180.26 | 89.24 |
| GTX 1070 | 44.86 | 42.12 | 102.37 | 54.29 | 41.64 |
| RTX 3080 | 12.61 | 11.41 | 40.93 | 16.04 | 11.37 |
| Accelerate rate | | | | | |
| GTX 1050 | 54.52 | 56.32 | 14.79 | 27.59 | 55.74 |
| GTX 1070 | 110.88 | 118.09 | 48.59 | 91.62 | 119.45 |
| RTX 3080 | 394.45 | 435.94 | 121.53 | 310.10 | 437.47 |

**Table 4.** *Cont.*

| Graphics Card | Version | | | | |
|---|---|---|---|---|---|
| | Replication | Replication (Shared Memory) | Transposition | Transposition (Shared Memory) | Transposition (Non-Bank Conflict) |
| | Bandwidth (GB/s) | | | | |
| GTX 1050 | 85.64 | 88.47 | 23.23 | 43.34 | 87.54 |
| GTX 1070 | 174.39 | 185.45 | 76.31 | 143.93 | 187.61 |
| RTX 3080 | 619.62 | 864.82 | 190.89 | 487.07 | 687.09 |

From the table, we can see that the use of shared memory has a certain speedup effect on both matrix replication and transpose operations. The operation speed of matrix replication is significantly faster than matrix transpose; this is because the data of the matrix copy process is continuous, and the data corresponds to the matrix position one by one. Matrix replication can make full use of the advantages of coalescing access and reduce the number of accesses, so matrix replication can reach a very high bandwidth of 619.62 GB/s (RTX3080), and it is improved to 864.82 GB/s by using shared memory.

For transposition, the operation of the data is skipped, so the data transfer cannot use coalescing access, which requires multiple fetching data from the global memory, and the bandwidth of the program is 190.89 GB/s (RTX3080). After using shared memory, the GPU computing bandwidth of GTX 1050 can reach 43.34 GB/s, which is increased by 86%. For GTX 1070, the bandwidth can increase by 88%, and for RTX 3080, the bandwidth can increase 155%. The higher the clock rate of the graphics card, the more obvious the bandwidth improvement effect.

It can be seen from the data in the table that the rate of transpose operation is still lower than that of matrix replication after using shared memory for both 10 series graphics cards and 30 series graphics cards. This indicates that its performance is not fully utilized in the matrix transpose operation due to the existence of bank conflicts. After the optimization of matrix transpose without bank conflict, the bandwidth can reach 99% of the matrix copy using shared memory, which shows that the data transmission capacity between shared memory and thread can be fully utilized.

This subsection shows that the computational efficiency of GPU parallelism is 122 times that of the CPU serial program. With shared memory, the ratio is increased to 310 times. At the same time, this subsection studies the use of the augmented matrix method to avoid bank conflicts, which further improves the rate by 41%, reaching 437.5 times that of serial calculation. Its advantages are that it is easy to modify and operate and has a good acceleration effect on the kernel. Its advantage is that the optimization method is simple and the effect of kernel acceleration is good. The only disadvantage is that it will occupy more shared memory.

## 4. Research on Asynchronous Parallelism in GPU Computing

An important concept in CUDA is a stream, which represents the execution queue of a series of instructions. GPU asynchronous parallelism is that the data and resources are divided into multiple parts, and the Host starts each execution sequence for the CUDA stream to process separately. Thus, it has the effect of overlapping data transmission and device calculation. Compared with thread parallelism, asynchronous parallelism using the CUDA stream is a higher level of parallelism. Thread parallelism is a level of parallelism for data, and streams are a level of parallelism for instructions.

*4.1. Comparison and Analysis of Different Asynchronous Parallel Methods*

For a subroutine that needs to be computed in parallel on the GPU, two types of operations (transfer of data between Host and Device and Kernel calculation) are required. Taking GTX 1050 as an example, this section visualizes the time of Memcpy D2H, kernel function and Memcpy H2D through the Nvprof tool, so as to analysis the time of the

CUDA stream under different conditions. According to the different tasks divided, GPU computing is divided into three versions.

Version 1: For computing tasks, one stream is used to realize the transmission of data from Host to Device, kernel execution, and then transferring data from Device to Host. The code is in Listing 2:

**Listing 2.** Code of Version 1.

```
1    !CPU Subroutine
2    Call Function_1 (a)
3
4    !GPU parallel Subroutine
5    a_d = a
6    call kernel <<<n/blockSize,blockSize>>>(a_d,0)
7    a = a_d
```

From code 1, compared to CPU serial computation, we can see that GPU parallel computation generates additional data transfer consumption (a_d = a,a = a_d). As can be seen from Figure 6, data transmission takes a lot of time; MemcpyH2D takes 1.2685 and MemcpyD2H takes 1.2581, accounting for 52.31% of the total computing time of version 1. This section investigates how to hide the time of data transmission: Version 2 and Version 3.

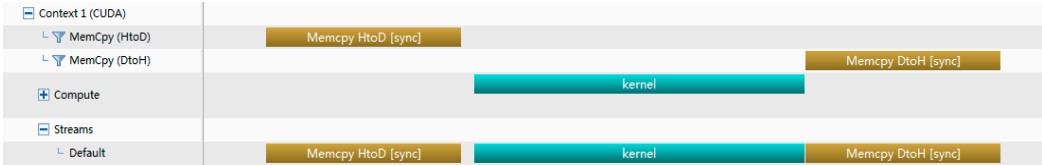

**Figure 6.** Time analysis of Version 1 (nstreams = 1, RTX 3080).

Version 2: The whole computing tasks are divided into several subtasks. Each CUDA stream completes the whole process of subtask. The code is in Listing 3:

**Listing 3.** Code of Version 2.

```
1    do i = 1,nStreams
2        offset =(i − 1)*streamSize
3        istat = cudaMemcpyAsync(a_d(offset + 1),a(offset + 1),streamSize,stream(i))
4        call kernel <<<streamSize/blockSize,blockSize,0,stream(i)>>>(a_d,offset)
5        istat = cudaMemcpyAsync(a(offset + 1),a_d(offset + 1),streamSize,stream(i))
6    end do
```

As shown in Figures 7 and 8 for Version 2, the object of the stream is the whole process of subtasks. The CUDA stream operates on the whole process, so the Memcpy H2D and Memcpy D2H cannot be parallel, as indicated by the red square in Figures 7 and 8. When nstreams is 4, the computation is split into 4 parts, we can see that Memcpy can overlap with the Kernel in three of the four parts, and the data transfer overlap rate is 75%. When nstreams is 10, the computation is split into 10 parts, we can see that 9 parts of Memcpy H2D can overlap with the Kernel and 6.75 parts of Memcpy D2H can overlap with the Kernel. The total overlap of data transmission is 79%.

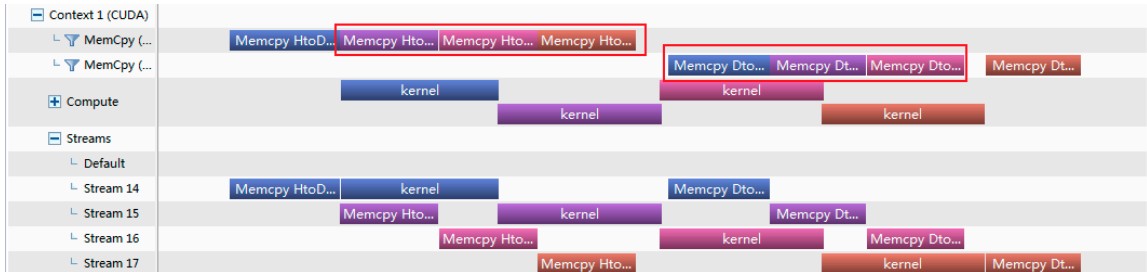

**Figure 7.** Time analysis of Version 2 (nstreams = 4, RTX 3080).

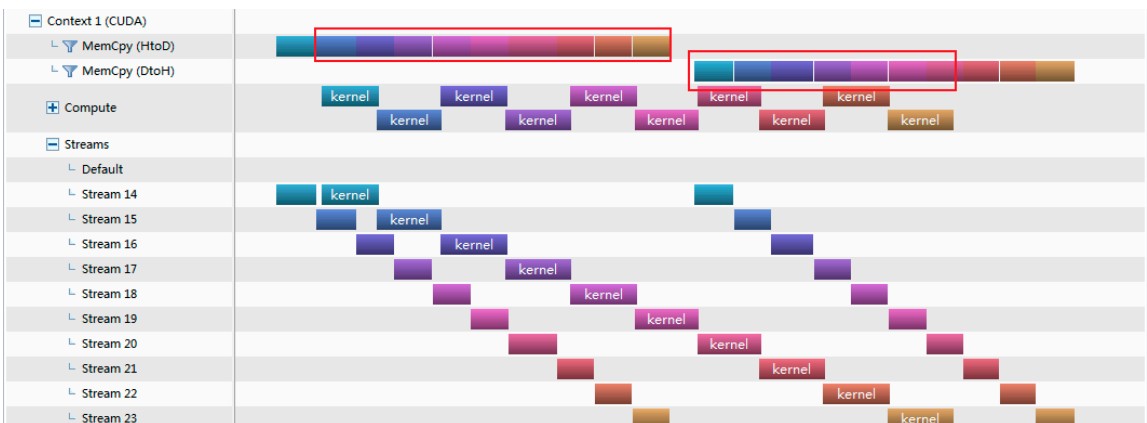

**Figure 8.** Time analysis of Version 2 (nstreams = 10, RTX 3080).

Version 3: The whole computing tasks are divided into three parts (Memcpy H2D, Kernel and Memcpy D2H). Then each part is divided into several subtasks to be executed by CUDA streams. The code is in Listing 4:

**Listing 4.** Code of Version 3.

```
1   do i = 1,nStreams
2       offset = (i − 1)* streamSize
3       istat = cudaMemcpyAsync(a_d(offset + 1),a(offset + 1),streamSize,stream(i))
4   end do
5   do i = 1,nStreams
6       offset = (i− 1)* streamSize
7       call kernel <<<streamSize/blockSize,blockSize,0,stream(i)>>>(a_d,offset)
8   end do
9   do i = 1,nStreams
10      offset = (i− 1)* streamSize
11      istat = cudaMemcpyAsync(a(offset+1),a_d(offset+1),streamSize,stream(i))
12  end do
```

As shown in Figures 9 and 10 for Version 3, the object of stream processing is the three parts (Memcpy H2D, Kernel and Memcpy D2H) that are divided. Thus, Memcpy D2H do not need to wait for all Memcpy H2D to complete before executing. For example, Memcpy D2H of Stream14 and Memcpy H2D of Stream16 in Figure 9 have a certain overlap time. When nstream is 4, although the overlap of memcpyD2H and Memcpy H2D saves some time, the time saved is wasted in waiting for kernel operation. Therefore, Version 2 and Version 3 take almost the same amount of time. When nstream is 10, whether MemcpyD2H or MemcpyH2D, 9 parts can overlap. The overlap rate of the total data transmission is 90%. Therefore, the time consumption of version 3 is lower than that of version 2.

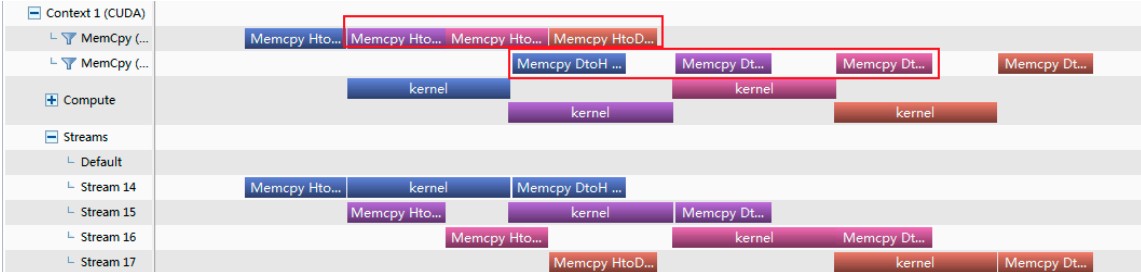

**Figure 9.** Time analysis of Version 3 (nstreams = 4, RTX 3080).

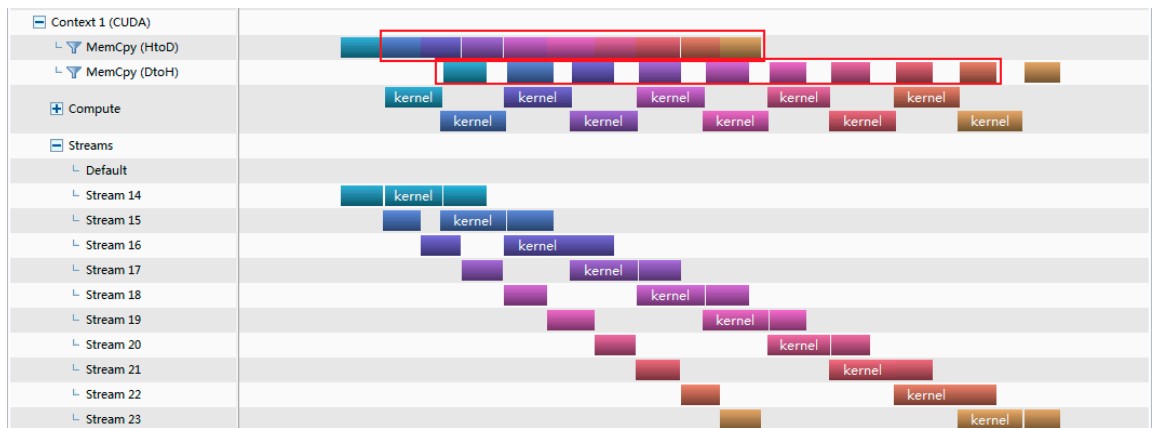

**Figure 10.** Time analysis of Version 3 (nstreams = 10, RTX 3080).

*4.2. Overlap Rate Theory of Memcpy*

In this paper, an asynchronous parallel overlap rate theory of Memcpy is proposed to explore the impact of different methods and different stream numbers on the overlap rate. The Memcpy version 1 cannot overlap, so it will not be discussed here.

1.    Overlap rate formula of Memcpy of version 2

For version 2, the time of Memcpy H2D and MemcpyD2H cannot overlap each other. Because the first and last Memcpy cannot be covered, when nstreams is $n$, the number of Memcpy that can be covered is $2(n-1)$. The overlap rate of Memcpy depends on whether the n kernel operations can cover the time of $2(n-1)$ Memcpy. The calculation time formula of version 2 is as follows:

$$max(nT_{kernel}, (n-1)(T_{H2D} + T_{D2H})) + (T_{H2D} + T_{D2H}) \tag{7}$$

where $T_{kernel} = T_{kernel}^{total}/n$, $T_{kernel}^{total}$ is total time of kernel execution, $T_{kernel}$ is kernel time of subtask. $T_{H2D}$ is the time of Memcpy H2D, $T_{D2H}$ is the operation time of MemcpyD2H, $n$ is the number of CUDA streams. After extracting the common factor:

$$max\left(\frac{nT_{kernel}}{(n-1)(T_{H2D} + T_{D2H})}, 1\right)(n-1)(T_{H2D} + T_{D2H}) + (T_{H2D} + T_{D2H}) \tag{8}$$

When $\frac{nT_{kernel}}{(n-1)(T_{H2D}+T_{D2H})}$ is greater than 1, the computation time is constrained by the Kernel execution time, and vice versa, the computation time is constrained by the Memcpy time. It can be seen from Table 1, $T_{H2D}$ and $T_{D2H}$ are approximately equal. For the Memcpy time to be covered, $T_{kernel}$ must satisfy $T_{kernel} > \frac{2(n-1)}{n}T_{H2D}$. Similarly, the number of fully covered Memcpy:

$$N = floor\left[min\left(\frac{nT_{kernel}}{(n-1)(T_{H2D} + T_{D2H})}, 1\right)\right](n-1) \tag{9}$$

2. Overlap rate formula of Memcpy of version 3

For version 3, when the program executes the second kernel operation, the output value of the first kernel executes the MemcpyD2H command, and the input value of the third kernel executes the Memcpy H2D command at the same time. There are three commands going on at the same time. Similarly, for a task divided into n parts, the time of Memcpy H2D and MemcpyD2H can overlap except for the first and last Memcpy, so the computation time after overlapping is determined by the time of n kernels and (n − 1) Memcpy. The calculation time formula of version 2 is as follows:

$$max\left(\frac{nT_{kernel}}{(n-1)T_{H2D}},1\right)(n-1)T_{H2D} + (T_{H2D} + T_{D2H}) \tag{10}$$

Similarly, for the Memcpy time to be covered, $T_{kernel}$ must satisfy $T_{kernel} > \frac{(n-1)}{n}T_{H2D}$. the number of fully covered Memcpy:

$$N = floor\left[min\left(\frac{nT_{kernel}}{(n-1)T_{H2D}},1\right)\right](n-1) \tag{11}$$

Comparing the calculation time formulas of version 2 and version 3, we find that when both of them are in the optimal condition, version 2 can overlap $(n-1)T_{H2D}$ more than version 3. This can show that version 3 is better than version 2 in theory. In the actual calculation process, we should analyze the relationship between Kernel and Memcpy, and select the appropriate number of CUDA streams according to $\frac{T_{kernel}}{T_{H2D}}$ to achieve the optimal working condition, as shown in Figure 11.

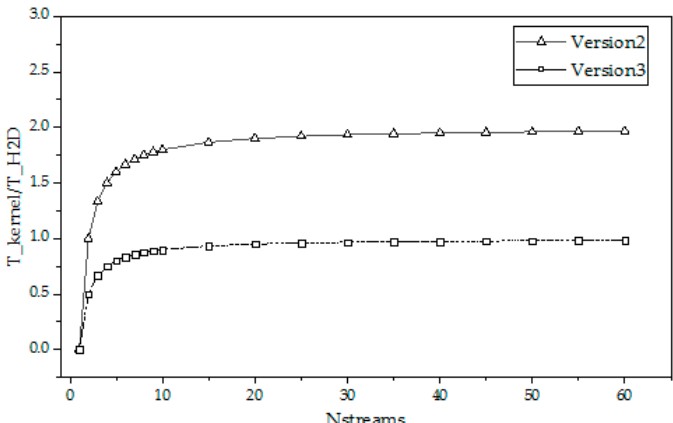

**Figure 11.** The lowest $\frac{T_{kernel}}{T_{H2D}}$ in the optimal working condition.

### 4.3. Analysis of Theoretical Operation Time and Practical Operation Time

Taking GTX 1050 as an example, it can be seen from nvprof that the execution time of the kernel function is 2.1325 ms, the execution time of MemcpyH2D is 1.2633 ms, and the ratio is 1.69. Figure 12 shows that version 2 can reach the optimal working condition when nstream is 7. As nstream becomes larger, $\frac{T_{kernel}}{T_{H2D}}$ cannot meet the requirements of the optimal operating condition, and Memcpy cannot achieve sufficient overlap. This is consistent with the phenomenon in Figure 12a that the actual overlap rate reaches the highest at nstream = 7. For version 3, $\frac{T_{kernel}}{T_{H2D}}$ is greater than 1, which meets the conditions of the optimal working condition. Figure 12b shows that the actual overlap rate and the theoretical overlap rate are consistent for any nstream number.

Through the comparison of the actual calculation time and the theoretical calculation time, although we can see that there is a certain gap between the two, it is very small and the change trend is consistent. The overlap rate of Memcpy can analyze the influence of the number of streams on the calculation time to a certain extent, and can be used to predict

the execution time of the task, which is of guiding significance for parallel compilation and optimization analysis of programs.

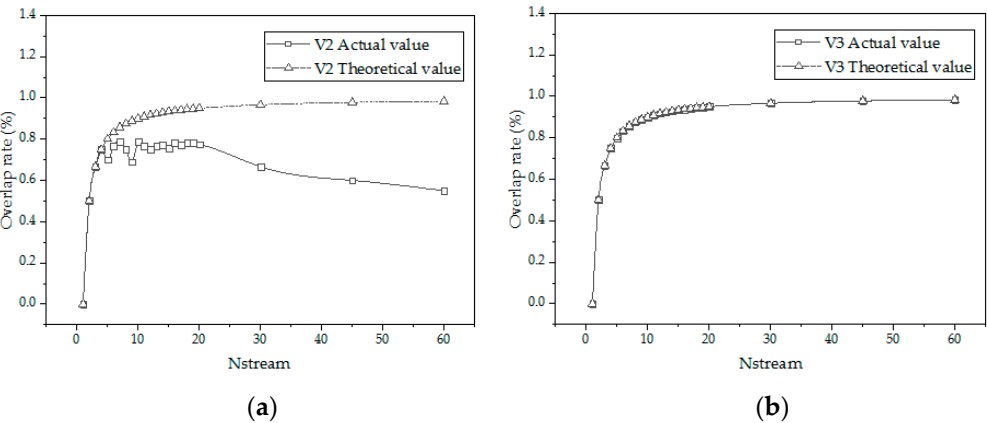

**Figure 12.** Actual and theoretical overlap rates (RTX 3080). (**a**) Version 2. (**b**) Version 3.

Under the condition of array size $4 \times 10^6$, the total time of CUDA stream operation is 4.971 ms, and the time of Memcpy(H2D) is 2.5266 ms. The theoretical computation time of the program can be calculated using Formulas (8) and (10), as shown in Figure 13.

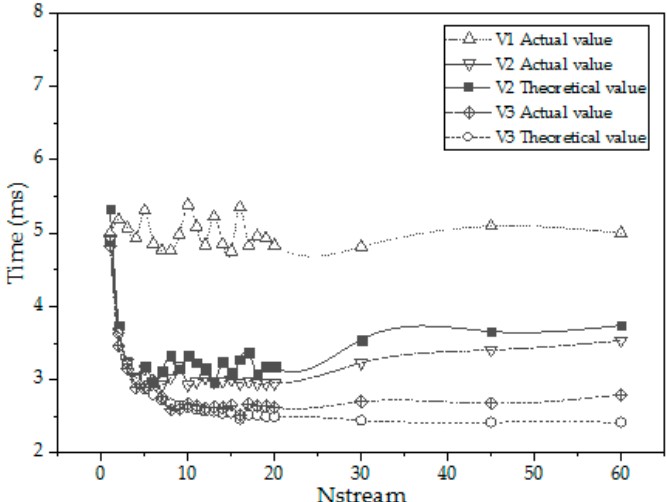

**Figure 13.** Actual and theoretical execution time (RTX 3080).

As shown in Figure 13, the actual calculation time is greater than the theoretical calculation time for both version 2 and version 3. This is because the use of CUDA streams not only brings the gain of Memcpy time overlap to reduce the execution time, but also GPU parallel computing itself comes with additional time consumption, as shown in the red box in Figure 14. These include the processes of cuCtxDetach, cuMemcpyH2Dasync, launch kernel, etc., of which cuMemcpyH2Dasync and launch kernel will increase with the increase in the number of nstreams.

Despite the consumption of other extra time, asynchronous parallelism is better than traditional parallelism, and Version 3 takes the least time. The parallel accelerate rate of each version is shown in Table 5. Version 2 can improve the speed by 65% on the basis of ordinary GPU parallelism, and Version 3 can improve 100%. Compared with serial calculation, the accelerate rates are 50.90 and 61.42.

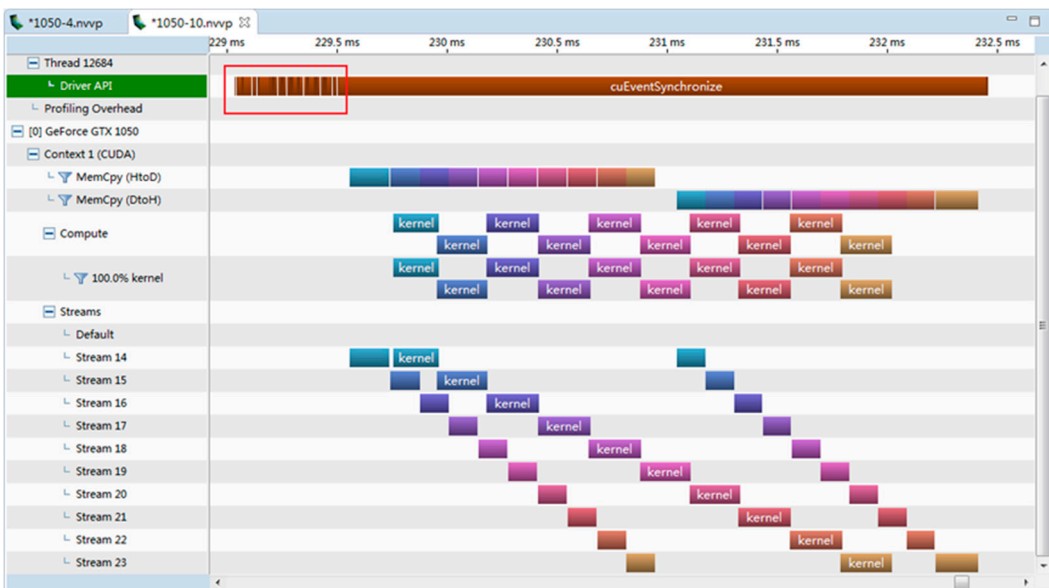

**Figure 14.** Whole time of Version 2 (nstreams = 10, RTX 1050).

**Table 5.** Accelerate ratio of each version.

| Version | Time (ms) | Accelerate Rate | Cumulative Accelerate Rate |
|---------|-----------|-----------------|----------------------------|
| Serial  | 148.63    | 1               | 1                          |
| V1      | 4.83      | 30.77           | 30.77                      |
| V2      | 2.92      | 1.65            | 50.90                      |
| V3      | 2.42      | 2.00            | 61.42                      |

### 4.4. Limitations of Parallel Algorithms

On the one hand, not all code is suitable for parallelization. When the amount of data to be transmitted is large and the calculation amount is relatively small, if parallel processing is adopted, the time consumption required to open up parallel fields and thread scheduling is even greater than the time consumption of computing itself, and the gain is not worth the loss.

On the other hand, not all loops can be parallelized. Before parallelizing loops, we must ensure that there is no data correlation (loop dependency or data competition) between the loops. When two threads operate on a variable at the same time and one of the operations is a write operation, the two threads have a data race. In this case, the data read out is not necessarily the data of the previous write operation, and the data written may not be required by the program.

For a loop with cyclic dependence, the calculation of the loop index J is related to the calculation result of the loop index I, or the loop index I iterates with itself, such as Listing 5. This loop can only be executed sequentially, otherwise the result of parallel execution will give the wrong result, as shown in Figure 15.

As can be seen from Figure 15, for a dependent loop, the value of a(i) is related to the value of a(i + 1). If the calculation is serial, the loop index i is executed sequentially. The assignment of a(i + 1) is after the assignment of a(i), so the change of a(i + 1) will not affect the assignment of a(i). However, in parallel computation, we cannot determine and predict the execution order of the loop index i. For example, in parallel computation of Test-1, a(9) = a(10) + c(9) executed by thread 2 precedes a(8) = a(9) + c(8) executed by thread 1. Therefore, in the process of execution, a(8) used a(9) after the assignment change, thus obtaining the wrong calculation result. At the same time, by comparing the parallel computation of Test-1 and Test-2, we can see that for the same calculator, the thread number of the execution of the loop index i on each execution cannot be determined. In addition,

the execution end times of different loop indexes i cannot be determined. Therefore, we cannot parallelize the code with loop dependence. When the program is reformed and optimized in parallel, the optimization effect of processing should be analyzed according to different situations.

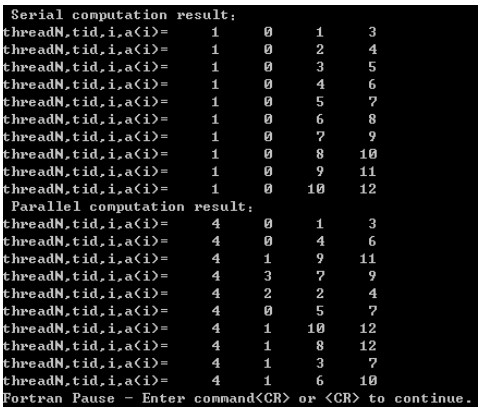

Test 1 of Loop dependence　　　　　　　　Test 2 of Loop dependence

**Figure 15.** Diagram of data errors caused by loop dependencies.

**Listing 5.** Code of loop dependencies.

```
1   do i = 1, n
2       a(i) = a(i + 1) + c(i)
3   end do
4   do i = 1, n
5       do j = 1, n
6   a(i,j) = a(i + 1,j + 1) + c(j)
7       end do
8   end do
```

## 5. Application of GPU Parallel Computing in Engineering

In the field of concrete temperature control simulation, many factors such as environmental conditions, materials and crack prevention measures should be considered in predicting cracks during construction. In order to obtain more accurate results, the element size of the finite element model is getting smaller and smaller, which leads to the increase of the calculation amount. In order to improve the calculation efficiency, GPU parallel is introduced into the temperature control simulation of massive concrete.

According to the GPU parallel optimization method proposed above, the Fortran finite element program for water pipe cooling temperature and stress field of our research group was reconstructed and optimized in parallel. After the program is parallelized, serial and parallel tests are carried out with an engineering example, and the calculation results of serial and parallel are compared.

### 5.1. Simulation Calculation Model and Material Parameters

According to the drawing of a project of a concrete gravity dam, the finite element model of 211,374 elements and 233,983 nodes was established to compare the efficiency and precision of parallel computing and serial computing. The finite element model is shown in Figure 16. In the simulation calculation of the temperature field, the surrounding and bottom surface of the foundation is the adiabatic boundary, and the upper surface is the heat dissipation boundary. The symmetry plane of the structure is the adiabatic boundary. The construction of a temporary seam surface, and the structure of a permanent seam surface, when not covered as the heat dissipation boundary, after covering is the adiabatic boundary. The other surfaces are heat dissipation boundaries. The foundation material below the floor of the structure is mainly andesitic breccia. C15

concrete is used for the central of the dam, C30 concrete is mainly used for the dam body and the overflow surface and other parts are made of C35 anti-impact and wear-resistant concrete. The thermal parameters of this paper are derived from the inversion of the measured temperature data of the project, and the mechanical parameters are derived from the laboratory mechanical tests.

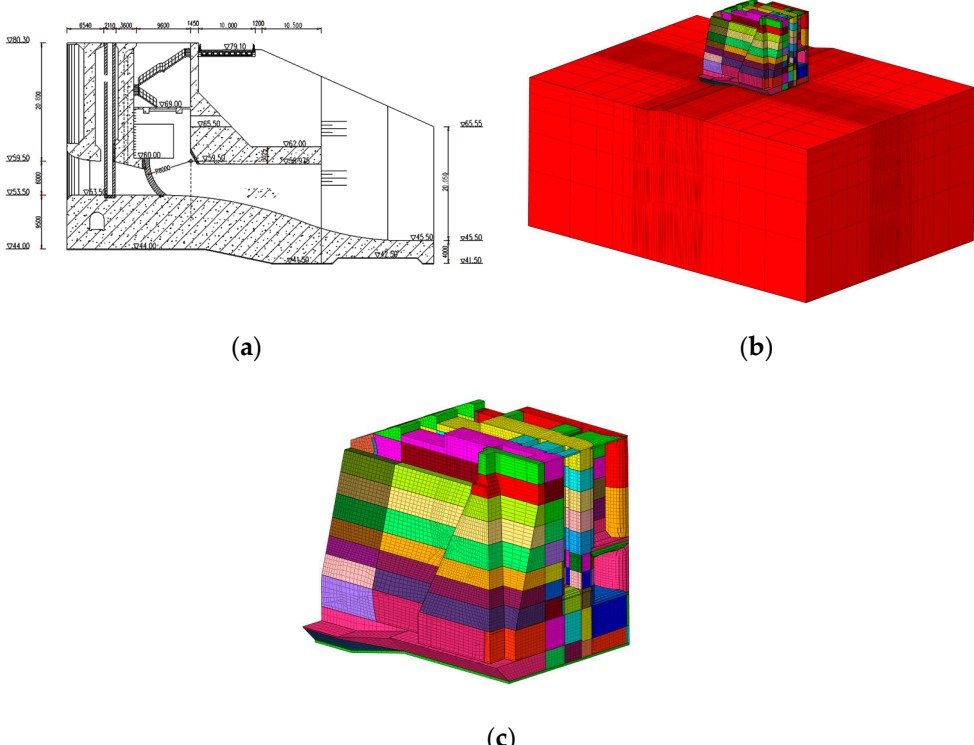

(**a**)

(**b**)

(**c**)

**Figure 16.** Finite element model of a project bottom hole section. (**a**) Vertical profile of dam bottom. (**b**) Integral finite element model. (**c**) Finite element model of dam section.

Calculation Condition: The concrete will be cooled by water from an early age with a long time and slow temperature drop rate. Moreover, enhancing surface insulation and a setting construction joint are applied.

The thermodynamic parameters of various materials are shown in Table 6. The temperature data of the dam site in recent years are analyzed. In order to facilitate calculation, the mean monthly temperature of many years is synthesized into a cosine curve.

$$T_\alpha(\tau) = 16 + 12.617 \times cos\left(\frac{\pi}{6} \times (\tau - 6.25)\right) \tag{12}$$

**Table 6.** Thermal and mechanical parameters of materials.

| Category | Thermal Convection (kJ/m·h °C) | Specific Heat (kJ/(kg·°C)) | Thermal Diffusivity (m²/h) | Linear Expansion Coefficient (×10⁻⁶/°C) | Poisson Ratio | Density (kg/m³) | Final Elasticity Modulus (GPa) | Final Value of Autogenous Volume Deformation (×10⁻⁶) | Final Value of Adiabatic Temperature Rise (°C) |
|---|---|---|---|---|---|---|---|---|---|
| BASE | 13.62 | 0.716 | 0.0076 | 8.0 | 0.240 | 2500 | 35.5 | — | — |
| C15 | 3.32 | 1.290 | 0.0011 | 8.0 | 0.167 | 2305 | 22.0 | 50 | 30.0 |
| C30 | 4.13 | 0.989 | 0.0017 | 8.7 | 0.167 | 2329 | 30.0 | 100 | 54.5 |
| C35 | 4.43 | 0.958 | 0.0019 | 9.0 | 0.167 | 2340 | 35.0 | 107 | 61.0 |

### 5.2. Experimental Platform Parameters

The debugging environment of the author's component program adopts the Windows operating system. A Cuda-based hybrid compiler language is implemented in the PGI Fortran compiler platform. The GPU used in the calculation is NVIDIA® GeForce RTX™

3080, and the CPU is Intel®Core™i7-12700k CPU @ 3.6 GHz. Details of the experimental platform are shown in Table 7.

**Table 7.** Experimental platform parameters.

| Operating System | Processor | Primary Frequency | Number of Cores | SM | Memory | Memory Copy (GiB/s) | |
|---|---|---|---|---|---|---|---|
| | | | | | | H2D | D2H |
| Windows10-64 bit | Intel i7-12700K | 3.6 GHz | 20 | - | 16 G | | - |
| | RTX3080 | 1.73 GHz | 8960 | 70 | 10 G | 23.97 | 23.71 |
| Compiling Environment | | Visual Studio 2010 | | | https://learn.microsoft.com/ | | |
| Compiler | | PGI Visual Fortran | | | https://www.pgroup.com/index.htm | | |
| Tool Kit | | CUDA Toolkit 10.0 | | | https://developer.nvidia.com/ | | |
| | | cuDNN | | | https://developer.nvidia.com/rdp | | |

Note: The website was accessed in 23 June 2022.

*5.3. Comparison of Calculation Results*

This section takes a concrete gravity dam as the research object, and simulates the temperature field and stress field of the concrete dam by using the GPU parallel algorithm proposed in this paper. The results of GPU parallel computation are compared with those of CPU, and the accuracy and efficiency of the GPU parallel algorithm are verified. Due to space constraints, we only select the middle section of the dam for temperature and stress description. The tensile stress is positive and the compressive stress is negative. The calculation results of temperature and stress are shown in Figure 17, and the calculation time is shown in Table 8.

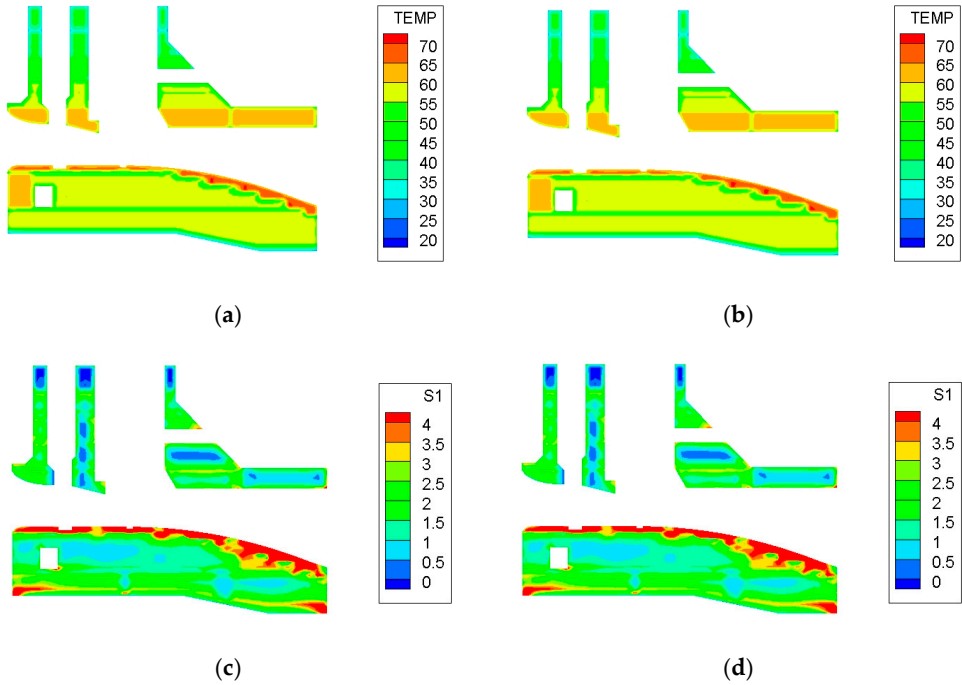

(a)

(b)

(c)

(d)

**Figure 17.** Temperature and stress envelope graphs for different calculation methods. (**a**) Temperature—CPU serial calculation. (**b**) Temperature—GPU parallel computation. (**c**) Stress—CPU serial calculation. (**d**) Stress—GPU serial calculation.

**Table 8.** Calculation time and accelerate rate of parallel.

| Type of Computation | Computing Time (s) | Accelerate Rate |
| --- | --- | --- |
| CPU serial | 7693 | — |
| CPU parallel | 5107 | 1.51 |
| GPU parallel | 2741 | 2.81 |

It can be seen from the above temperature and stress envelope diagrams that the results of GPU parallel calculation are basically consistent with those of CPU serial calculation, indicating that the accuracy of the optimized GPU parallel algorithm meets the needs of engineering calculation.

It can be seen from the Table 8, compared with serial computing, the total time of CPU parallel computing and GPU parallel computing decreased by 33.61% and 64.37%. GPU parallel computing significantly shortened the total time of computing. GPU parallel computation can effectively improve the efficiency of the concrete temperature control simulation program, so that the simulation program can better serve the prediction of concrete cracks during the construction period. Therefore, more reasonable temperature control and crack prevention measures are taken to protect the safety of the concrete structure. At the same time, we can adopt the principle and method of GPU parallelization to analyze and transform other computing programs in parallel. The theory of GPU parallel computing is universal for code that can be parallelized, but needs to be adjusted for its own characteristics (amount of data, amount of computation, number of iterations, etc.).

There is a certain gap between the acceleration efficiency of the overall program and the GPU acceleration ratio shown by shared memory optimization and asynchronous parallelism. Because on the one hand, there is code that can only be executed in serial in the program, which cannot be modified in parallel, and on the other hand, for the subroutine with a small amount of computation, parallelization will bring a large additional cost, which outweighs the gain. After adopting the parallel transformation and optimization method described in this paper, in the case of ensuring the accuracy of calculation, the GPU computing acceleration ratio of the whole program reaches 2.81 times.

The program is optimized for shared memory and CUDA stream's hiding of memory access time, and the computing programs in other fields are not studied. Because each application targets different fields and solves different problems, it is impossible to give a detailed classification. At the same time, the performance of different models of GPU processors, the number of nested layers per Do loop, the number of loops, the amount of data, the amount of computation and the data type are different, which will affect the additional overhead of the program. Therefore, it is necessary for the programmer to make adjustments according to the characteristics of their own program under the guidance of GPU parallel computing theory and methods. And as mentioned above, the additional cost of parallel computing involves a wide range of factors. Therefore, it is of great significance to further study the effect of extra cost on the efficiency of parallel computing, and it is a worthy research direction.

## 6. Conclusions

Aiming at the problem that the current conventional serial computing efficiency is low and cannot meet the engineering requirements, GPU parallel computing is introduced into the large massive concrete temperature control simulation calculation program. An improved analytical formula for GPU parallel algorithms is proposed; it makes up for the shortcomings of the traditional algorithms that only focuses on time, which is conducive to finding out the direction of program optimization. Optimization of the parallel program is studied through two aspects: shared memory and CUDA stream. The optimized program obtains a better speedup ratio.

1. From the improved analytical formula, GPU parallel programs should be optimized from the following aspects: Hardware level: replace the GPU with stronger perfor-

mance to obtain more threads and higher clock rate. Algorithm level: modify the algorithm to increase the proportion of parallel operations, improve the running efficiency of kernel functions and overlap more data transmission time;

2. The data access mode of parallel programs is optimized by using shared memory, and the problem of bank conflicts is further solved. For matrix transpose operations of finite element operations, 437.5× acceleration is achieved;

3. This paper implements asynchronous parallelism on the GPU through CUDA streams, which can hide the time of data access. Overlap rate theory of Memcpy is proposed to guide the optimization of asynchronous parallel programs. For GPU kernel subroutines of matrix inner products, compared with ordinary GPU parallel programs that do not use asynchronous parallelism, it can achieve nearly twice the acceleration. Compared with serial programs, it can achieve 61.42× acceleration. Not all programs are suitable for parallelization and need to be analyzed on a case-by-case basis;

4. The Fortran finite element program for temperature and stress fields of concrete is reconstructed and optimized in parallel. The GPU parallelization of the program plays a role in improving computational efficiency while ensuring the computational accuracy.

**Author Contributions:** Conceptualization, X.Z., J.J. and S.Q.; methodology, X.Z., Y.W. and S.Q.; software, X.Z., Y.W. and S.Q.; formal analysis, X.Z., Y.W. and S.Q.; investigation, X.Z. and M.Y.; resources, X.Z. and J.J.; data curation, X.Z., J.J. and M.Y.; writing—original draft preparation, X.Z. and J.J.; writing—review and editing, X.Z., Y.W. and S.Q.; visualization, X.Z., Y.W. and S.Q. supervision, Y.W. and S.Q.; project administration, Y.W. and S.Q.; funding acquisition, X.Z. and S.Q. All authors have read and agreed to the published version of the manuscript.

**Funding:** This research was funded by the National Natural Science Foundation of China, grant number 52079049, Water Conservancy Science and Technology Project of Henan Province, China in 2022.

**Data Availability Statement:** Data are contained within the article.

**Acknowledgments:** The authors acknowledge the Fundamental Research Funds for the Central Universities and the Postgraduate Research & Practice Innovation Program of Jiangsu Province.

**Conflicts of Interest:** Authors Jiping Jin and Yajun Wang were employed by the company The First Engineering Bureau of Henan Water Conservancy. The remaining authors declare that the research was conducted in the absence of any commercial or financial relationships that could be construed as a potential conflict of interest.

## Abbreviations

The following abbreviations are used in this manuscript:

| | |
|---|---|
| CUDA | Compute Unified Device Architecture |
| CPU | Central Processing Unit |
| GPU | Graphics Processing Unit |
| H2D | Host to Device |
| D2H | Device to Host |
| Dg/Db | Dim of Girde/ Dim of Block |
| Memcpy | Memory Copy |

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
