# Peer review of "Research on the Application and Performance Optimization of GPU Parallel Computing in Concrete Temperature Control Simulation"

_buildings, doi:10.3390/buildings13102657_

Round 1
Reviewer 1 Report
The title of the article shows it is applying the GPU computing on concrete temperature control simulation. The introduction also claims that. "The GPU parallel program compiled and optimized by CUDA Fortran platform can effectively improve the computational efficiency of the simulation program for concrete temperature control, and better serve for engineering computing". But in between the paper diverts to finding the parallelization algorithm of GPU threads instead of talking about any civil engineering problem. Therefore, this paper is not at all suitable with this title and out of the scope of this journal.
Author Response
Dear Editors and Reviewers,
On behalf of my co-authors, we thank you very much for giving us an opportunity to revise our manuscript, and we appreciate editors and reviewers very much for their positive and constructive comments and suggestions on our manuscript entitled “Research on the Application and Performance Optimization of GPU Parallel Computing in Concrete Temperature Control Simulation” (ID: buildings-2606605). Those comments are all valuable and very helpful for revising and improving our paper, as well as the important guiding significance to our researches. We have tried our best to revise our manuscript according to the comments. We would like to submit for your kind consideration. Revisions are marked in yellow in the revised manuscript. The responses to the reviewers’ comments are as following, if you have any more questions, we would be happy to address them, looking forward to hearing from you.
Thank you and best regards.
Yours sincerely,
Xuerui Zheng, Jiping Jin, Yajun Wang, Min Yuan and Sheng Qiang
Response to Reviewer 1 Comments
Point 1: The title of the article shows it is applying the GPU computing on concrete temperature control simulation. The introduction also claims that. "The GPU parallel program compiled and optimized by CUDA Fortran platform can effectively improve the computational efficiency of the simulation program for concrete temperature control, and better serve for engineering computing". But in between the paper diverts to finding the parallelization algorithm of GPU threads instead of talking about any civil engineering problem. Therefore, this paper is not at all suitable with this title and out of the scope of this journal.
Response 1: Firstly, thank you very much for pointing this out. This opinion has played a key role in my elaboration of this manuscript
In view of your valuable suggestions, we have added a new section 5 to explain the applicability of GPU parallelization transformation to the massive concrete temperature control simulation calculation program. In this section, the GPU parallel algorithm proposed in this paper is used to simulate the temperature field and stress field of concrete gravity dam. The results of GPU parallel computation and CPU parallel computation are analyzed, and the computation time of CPU serial computation, CPU parallel computation and GPU parallel computation is compared under the condition of ensuring the same accuracy
Many subroutines in this program can only be executed in serial, and cannot be parallelized transformation, so the acceleration ratio does not reach the acceleration ratio of shared memory optimization in section 3 and asynchronous parallel optimization in section 4. But in general, GPU parallelism is more efficient than CPU serial and CPU parallelism, The GPU parallelization of the program plays a role in improving computational efficiency while ensuring the computational accuracy.
In addition, We added two related references in line 63 an line 70 to better illustrate the research progress of parallel computing.

Reviewer 2 Report
Overall, this manuscript is a good starting point for research on the optimization of GPU parallel programs. However, it needs to be significantly improved before it can be published.
Here are some specific suggestions for improvement:
1. The article should provide more details about the improved analytical formula for GPU parallel algorithms. The formula is presented in a very concise way, and it is not clear how it is derived or how it can be used to optimize parallel programs.
2. The article should provide more experimental results to support the claims made about the effectiveness of the proposed optimizations. The results that are presented are limited to a few specific cases, and it is not clear how they generalize to other programs.
3.The article should discuss the limitations and trade-offs of the proposed optimizations. For example, the asynchronous parallelism approach may not be suitable for all programs, and it may introduce additional overhead.
The article is written in a technical style and may be difficult to understand for readers who are not familiar with GPU programming. It would be helpful to provide more explanations and examples to make the article more accessible to a wider audience.
Author Response
Dear Editors and Reviewers,
On behalf of my co-authors, we thank you very much for giving us an opportunity to revise our manuscript, and we appreciate editors and reviewers very much for their positive and constructive comments and suggestions on our manuscript entitled “Research on the Application and Performance Optimization of GPU Parallel Computing in Concrete Temperature Control Simulation” (ID: buildings-2606605). Those comments are all valuable and very helpful for revising and improving our paper, as well as the important guiding significance to our researches. We have tried our best to revise our manuscript according to the comments. We would like to submit for your kind consideration. Revisions are marked in yellow in the revised manuscript. The responses to the reviewers’ comments are as following, if you have any more questions, we would be happy to address them, looking forward to hearing from you.
Thank you and best regards.
Yours sincerely,
Xuerui Zheng, Jiping Jin, Yajun Wang, Min Yuan and Sheng Qiang
Response to Reviewer 2 Comments
Point 1: The article should provide more details about the improved analytical formula for GPU parallel algorithms. The formula is presented in a very concise way, and it is not clear how it is derived or how it can be used to optimize parallel programs.
Response 1: Firstly, thank you very much for pointing this out. We have added the content of the formula and the derivation process in Equation 4 and Equation 6, and explained the derivation process of the formula(marked in yellow in the line150-166)
We have further expounded and explained the meaning expressed by the formula(marked in yellow in the line184-197).The formula provides a more intuitive guide to improving the program. Through the formula, we can let the programmer understand more intuitively, in order to improve the acceleration ratio of GPU parallel programs, we need to pay attention to the following aspects. On the one hand, we can improve the GPU computing power through the hardware level, increasing the frequency and number of GPU cores. On the other hand, we can increase the scope of the parallel domain or reduce the additional consumption caused by parallel programs through the optimization of programs and algorithms. This manuscript mainly focuses on the additional consumption.
Point 2: The article should provide more experimental results to support the claims made about the effectiveness of the proposed optimizations. The results that are presented are limited to a few specific cases, and it is not clear how they generalize to other programs.
Response 2: Thank you for pointing this out. In fact, GPU parallel modification of serial programs is trivial and complex. And there is no way to provide a generalised and accurate approach to different codes. Because GPU models, computing platforms, loop nesting methods, computing volume size, and complexity of operations will affect Kernel execution configuration and the selection of parallel optimization methods. Due to the limited space of this paper, we discuss the idea of optimization, so only partial examples are given for each optimization method.
In view of your valuable suggestions, we have added a new section 5 to explain the applicability of GPU parallelization transformation to the massive concrete temperature control simulation calculation program. Many subroutines in this program can only be executed in serial, and cannot be parallelized transformation, so the acceleration ratio does not reach the acceleration ratio of shared memory optimization in section 3 and asynchronous parallel optimization in section 4. But in general, GPU parallelism is more efficient than CPU serial and CPU parallelism, The GPU parallelization of the program plays a role in improving computational efficiency while ensuring the computational accuracy.
Point 3: The article should discuss the limitations and trade-offs of the proposed optimizations. For example, the asynchronous parallelism approach may not be suitable for all programs, and it may introduce additional overhead.
Response 3: Firstly, thank you very much for pointing this out. Your opinion makes my article convincing.
By contrast with serial programs, we add a description of the additional cost of GPU parallel data transfer in line 349-354 of this manuscrip. We explain that the research content of this section is how to effectively hide the extra consumption brought by data transmission.
On line 446-452of the manuscript, we have added an overview of the additional costs associated with GPU parallel computing. The additional cost of GPU parallel programs comes from multiple sources. For example, parallel domain opening, Kernel execution configuration, bank cnflict, data transmission and so on. This manuscript mainly studies the additional cost of data transmission, and other additional costs are not described in detail due to space reasons.
We added section 4.4 to discuss that not all programs and code are suitable for GPU asynchronous parallelism. And it is reflected in the conclusion
In addition, We added two related references in line 63 an line 70 to better illustrate the research progress of parallel computing.

Reviewer 3 Report
Currently, there are higher requirements for the accuracy and scope of simulation calculations. Research on the application and performance optimization of GPU parallel computing in the simulation of concrete temperature control is the content of this paper. The article is extensive, yet interesting for the reader.
Reducing the calculation time of the temperature control simulation program has an important technical significance for the simulation of the temperature field. An improved GPU parallel algorithm analysis indicator formula is proposed here. It replaces the shortcomings of the traditional formula that focuses only on time. This article studies the optimal kernel execution configuration.
The present study further proposed a theoretical data access overlap rate formula to guide the selection of the number of streams to achieve optimal computing conditions. The parallel program compiled and optimized by the CUDA Fortran platform can effectively improve the computational efficiency of the simulation program for specific temperature control and better serve engineering calculations.
The post has chapters:
1. Introduction
2. Improved analytical formula for GPU parallel algorithms
3. Research on GPU Memory access optimization by using shared memory results
4. Research on asynchronous parallelism in GPU computing
5. Conclusions
Compared to serial programs, the new program can achieve an acceleration of about 60 times.
The article uses 37 validated publication sources.
It is interesting, easy to read, attractive to the reader.
I recommend publishing it as is.
Author Response
Dear Editors and Reviewers,
On behalf of my co-authors, we thank you very much for reviewing our manuscript, and we appreciate editors and reviewers very much for your approval of the content of our manuscript entitled “Research on the Application and Performance Optimization of GPU Parallel Computing in Concrete Temperature Control Simulation” (ID: buildings-2606605). On this basis, we further modify the content of our manuscript, trying to better present our research. Revisions are marked in yellow in the revised manuscript. If you have any more questions, we would be happy to address them, looking forward to hearing from you.
Thank you and best regards.
Yours sincerely,
Xuerui Zheng, Jiping Jin, Yajun Wang, Min Yuan and Sheng Qiang

Reviewer 4 Report
I have no comments on the article.
Author Response

(The authors gave the same response as above.)

Round 2
Reviewer 1 Report
Still this paper is not aligned with the scope of this journal. This can be submitted some of the computing journals like computation. Alternatively, more details of section 5 can be given here so that it will be well aligned with the scope of this journal
No comments
Author Response
Dear Editors and Reviewers,
On behalf of my co-authors, we thank you very much for giving us an opportunity to revise our manuscript, and we appreciate editors and reviewers very much for their positive and constructive comments and suggestions on our manuscript entitled “Research on the Application and Performance Optimization of GPU Parallel Computing in Concrete Temperature Control Simulation” (ID: buildings-2606605). Thank you and best regards. Those comments are all valuable and very helpful for revising and improving our paper, as well as the important guiding significance to our researches. We have tried our best to revise our manuscript according to the comments. We would like to submit for your kind consideration. Revisions are marked in yellow in the revised manuscript. The responses to the reviewers’ comments are as following, if you have any more questions, we would be happy to address them, looking forward to hearing from you.
Yours sincerely,
Xuerui Zheng, Jiping Jin, Yajun Wang, Min Yuan and Sheng Qiang
Response to Reviewer 1 Comments
Point 1: Still this paper is not aligned with the scope of this journal. This can be submitted some of the computing journals like computation. Alternatively, more details of section 5 can be given here so that it will be well aligned with the scope of this journal
Response 1: Firstly, thank you very much for pointing this out. In fact, I submit this manuscript to a special issue named Application of Computer Technology in Buildings, which is affiliated with journal Buildings. Whether it is a building or a concrete engineering structure, they are all buildings. Our research focuses on concrete structures, but also applies to concrete civil buildings. This manuscript aims to study how to use high performance computing method to improve the efficiency of engineering calculation and better serve the design and construction of engineering, which is of great significance to the design and construction of buildings. Especially for the mass concrete structure, its structure is complex, the finite element calculation is large, need to use high performance computing to improve the efficiency.I think the study of the manuscript fits the theme of the special issue.
In order to solve the problem of low efficiency of conventional serial computing and unable to meet engineering requirements, GPU parallel computing is used to mprove the efficiency of engineering calculations. We studied the application and optimization of GPU parallel computing from the aspects of GPU parallel algorithm analytic formula, shared memory and CUDA stream, and uses it in the temperature control simulation of mass concrete. The calculation efficiency of the program is improved.I think my article basically meets the requirements of the special issue for the writing direction. However, in order to be better aligned with the scope of Special Issue. I have made some amendments in section 5. Revisions are marked in yellow in the revised manuscript.
In section 5. we added the introduction of the temperature control simulation calculation program, and added the structure diagram of the dam body, and expounded the method of obtaining the ambient temperature. Modify table data to make significant numbers consistent. At the same time, the conclusion of this section has been modified.

Reviewer 2 Report
This paper presents a well-written and informative study on the use of GPU parallel computing to improve the performance of concrete temperature control simulation programs. The authors propose a number of optimizations, including an improved analytical formula for GPU parallel algorithm analysis, the use of shared memory, and asynchronous parallelism. They demonstrate the effectiveness of their proposed optimizations through experimental results, which show that they can achieve significant speedups over serial programs.
Here are some specific comments:
- The introduction is clear and concise, and it provides a good overview of the paper's contributions. The authors could add a brief discussion of the challenges of using GPU parallel computing in concrete temperature control simulation. For example, they could mention the need to deal with large amounts of data and the complex nature of the simulation problem. The paper could also be made more accessible to a wider audience by providing more explanations and examples in the introduction and discussion sections.
- The literature review is comprehensive and up-to-date, and it clearly demonstrates the need for the proposed research.
- The methodology section is well-written and provides sufficient detail about the experimental setup and procedures.
- The results section is clear and well-organized, and it presents convincing evidence of the effectiveness of the proposed optimizations.
- The discussion section is well-written and provides a good overview of the implications of the research.
- The paper could be improved by including more discussion of the following topics: The applicability of the proposed optimizations to other applications besides concrete temperature control simulation; The cost of the proposed optimizations, and the trade-off between performance improvement and cost.
- In the methodology section, the authors could provide more details about the hardware and software platform used for their experiments.
- In the results section, the authors could include a table or figure that summarizes the speedups achieved by their proposed optimizations.
- In the discussion section, the authors could discuss the limitations of their work and suggest directions for future research. For example, they could discuss the need to develop more general-purpose GPU parallel computing libraries for concrete temperature control simulation.
- The conclusions section is clear and concise, and it summarizes the main contributions of the paper.
The language is fine.
Author Response
Dear Editors and Reviewers,
On behalf of my co-authors, we thank you very much for giving us an opportunity to revise our manuscript, and we appreciate editors and reviewers very much for their positive and constructive comments and suggestions on our manuscript entitled “Research on the Application and Performance Optimization of GPU Parallel Computing in Concrete Temperature Control Simulation” (ID: buildings-2606605). We discussed the direction of improvement given by the reviewers, and revised and supplemented the suggestions in the manuscript. Revisions are marked in yellow in the revised manuscript. The responses to the reviewers’ comments are as following, if you have any more questions, we would be happy to address them, looking forward to hearing from you.
Thank you and best regards.
Yours sincerely,
Xuerui Zheng, Jiping Jin, Yajun Wang, Min Yuan and Sheng Qiang
Response to Reviewer 2 Comments
Point 1: The applicability of the proposed optimizations to other applications besides concrete temperature control simulation. The cost of the proposed optimizations, and the trade-off between performance improvement and cost. In the discussion section, the authors could discuss the limitations of their work and suggest directions for future research. For example, they could discuss the need to develop more general-purpose GPU parallel computing libraries for concrete temperature control simulation.
Response 1: Firstly, thank you very much for this suggest. We have added a brief introduction to GPU parallel computing applied to other programs in section 5 in page19 (marked in yellow).
Because each application targets different fields and solves different problems, it is impossible to give a detailed classification. At the same time, the performance of different models of GPU processors, the number of nested layers per Do loop, the number of loops, the amount of data, the amount of computation, and the data type are different, which will affect the additional overhead of the program.Therefore, it is necessary for the programmer to make adjustments according to the characteristics of their own program under the guidance of GPU parallel computing theory and methods. In order to avoid misleading readers, this manuscript does not give too much description, and will be presented in the author's next research.
Because of the above reasons, the additional cost of the program needs to be considered by many factors, which is a complex research direction, but also a very worthy research direction. Suggestions of the research direction is proposed at the end of section 5 in th manuscript.
Point 2: In the methodology section, the authors could provide more details about the hardware and software platform used for their experiments.
Response 2: Thank you for pointing this out. I have added more detailed information in Table 1 and Table 7 of the manuscript
Point 3: In the results section, the authors could include a table or figure that summarizes the speedups achieved by their proposed optimizations.
Response 3: Firstly, thank you very much for pointing this out. We made a revision in response to your comments. In Table 4 of section 4, we add data on acceleration ratios to make the results more convincing.
The results of the acceleration ratio are presented in section 3、section 4 and section 5. Because sections 3 and 4 are for the acceleration effect of different methods, section 5 is for the acceleration effect of the entire concrete temperature control simulation program, all of which do not belong to the same optimization type. Therefore, its acceleration effect was not integrated in the conclusion.
